

# Protein signatures linking history of miscarriages and metabolic syndrome: a proteomic study among North Indian women

Saurabh Sharma[1,*], Suniti Yadav[2,*], Ketaki Chandiok[2], Radhey Shyam Sharma[1], Vandana Mishra[1] and Kallur Nava Saraswathy[2]

[1] Bioresources & Environmental Biotechnology Laboratory, Department of Environmental Studies, University of Delhi, Delhi, India
[2] Molecular Anthropology Laboratory, Department of Anthropology, University of Delhi, Delhi, India
* These authors contributed equally to this work.

Corresponding authors
Vandana Mishra,
mistletoe_h@hotmail.com
Kallur Nava Saraswathy,
knsaraswathy@yahoo.com

## ABSTRACT

**Background:** Metabolic syndrome (MeS), a constellation of metabolic adversities, and history of miscarriage make women at a higher risk for cardiovascular diseases (CVDs). However, molecular evidence indicating a link between the two phenotypes (history of miscarriage and MeS) among women would offer an opportunity to predict the risk factor for CVDs at an early stage. Thus, the present retrospective study attempts to identify the proteins signatures (if any) to understand the connection between the history of miscarriage and MeS.

**Methods:** Age-matched 80 pre-menopausal women who were not on any medical intervention or drugs were recruited from a Mendelian population of the same gene pool. Recruited women were classified into four groups—(a) Group A—absolute cases with history of miscarriage and MeS, (b) Group B—absolute controls without any history of miscarriage and MeS, (c) Group C—cases with MeS but lack any history of miscarriage, (d) Group D—cases with history of miscarriage but lack MeS. Differentially expressed proteins in plasma samples of women from four groups were identified using 2-D gel electrophoresis and mass spectrometry.

**Results:** Three case groups (A, C, and D) showed 18 differentially expressed proteins. Nearly 60% of proteins (11/18) were commonly dysregulated in Group C (only with MeS) and Group D (only with miscarriage history). Nearly 40% of proteins (7/18) were commonly dysregulated in the three case groups (Groups A, C, and D), indicating a shared pathophysiology. Four proteins were exclusive but shared by case groups C and D indicating the independent routes for CVDs through MeS or miscarriages. In absolute cases, transthyretin (TTR) showed exclusive upregulation, which was further validated by Western blotting and ELISA. Networking analyses showed the strong association of TTR with haptoglobin, transferrin and ApoA1 hinting toward a cross-talk among these proteins which could be a cause or an effect of TTR upregulation.

**Conclusion:** The study provides evidence for molecular link between the history of miscarriage and MeS through a putative role of TTR. However, longitudinal

follow-up studies with larger sample size would further help to demonstrate the significance of TTR and other targeted proteins in risk stratification and the onset of CVDs.

# INTRODUCTION

Metabolic syndrome (MeS) shows a constellation of metabolic adversities, such as central obesity, high glucose, high blood pressure, high triglycerides, and low high-density lipoprotein cholesterol (*Grundy, 2005*). Individuals with MeS become susceptible to develop cardiovascular diseases (CVDs) and diabetes and such individuals show high mortality compared with those without MeS (*Xu et al., 2013*). The global trends also confirm that with the increase in MeS the adults also show a parallel increase in risk for CVDs, type-2 diabetes, and hypertension (*Ranasinghe et al., 2017*). MeS is more prevalent among women as compared with men (*Gu et al., 2005*; *Xi et al., 2013*; *Lim et al., 2011*; *Yeh, Chang & Pan, 2011*). Likewise, mortality due to CVD's is reportedly higher among women in contrast to men, with distinct characteristics linked to reproductive history among women (*Ranthe et al., 2013*).

It has generally been believed that women have a degree of protection against certain CVDs (such as atherosclerosis) in comparison with men (*Sharma & Gulati, 2013*); however, such a difference is attenuated in women with a history of adverse pregnancy events owing to underlying mechanisms that share the etiology of CVDs (*Maino et al., 2016*). For example, preeclampsia, stillbirth, and preterm birth exert similar effects on the endothelium and vascular system despite different etiologies. Atherosclerotic changes and inflammation in the vascular bed link adverse pregnancy and cardiovascular events (*Sharma & Gulati, 2013*); therefore, women with a history of pregnancy complications may develop a higher risk for different CVDs in later life (coronary heart disease, ischemic heart disease, myocardial infarction, etc.) (*Germain et al., 2007*; *McDonald et al., 2008*; *Oliver-Williams et al., 2013*; *Parker et al., 2014*). Some studies also suggest that woman after each additional miscarriage becomes more susceptible to myocardial infarction, cerebral infarction and renovascular hypertension as compared with women without a history of miscarriage (*Sharma & Gulati, 2013*). Pregnancy loss-induced inflammatory alterations (metabolic, hormonal, and hemostatic) may contribute to the development of CVDs suggesting a shared pathological mechanism (*Kharazmi et al., 2011*; *Fraser et al., 2012*; *Xu et al., 2013*), however the molecular link between both the conditions is not yet well established.

Human diseases are often treated separately, but they are mostly associated because of physiological interactions among common proteins, metabolic relationships, or interaction between signaling pathways. Analysis of common proteins among exclusive cases of two independent diseases and its comparison with the cases with co-occurrence

of both the diseases has been central to decipher the finer molecular links between the two metabolic states. Thus, it is important to investigate the molecular signatures that can link history of miscarriage and MeS for risk stratification in women for CVDs. Women with a history of miscarriage(s) but at a pre-CVD condition, that is, MeS form an ideal population for early therapeutic intervention for reducing the risk to the onset of CVDs (*Grundy, 2005*). However, epidemiological studies associating miscarriages to the elevated risk of MeS have not yet been conducted. A cohort study to identify dysregulated proteins in plasma of women with either of the conditions (MeS or miscarriage) or both miscarriage and MeS as compared with the absolute controls is a prerequisite to discover highly predictive biomarkers of CVDs for improving disease stratification methods and developing intervention or prevention methods for CVDs.

Thus, the present study was undertaken as a retrospective study to identify differences in the plasma protein expression patterns among women with a history of miscarriage and MeS to identify the biomolecular links between the two health conditions. The protein signature linking the two health conditions (history of pregnancy complications and MeS) offers an opportunity to predict the risk for CVDs among women at an early stage.

## MATERIALS AND METHODS

### Ethics statement

The study was approved by Ethics Committee, Department of Anthropology, University of Delhi, New Delhi, India (Ref. No. Anth/2010/455/1). All participants provided pre-informed written consent to participate in this study.

### Participants and plasma sample preparation

Fieldwork was carried out in 15 different villages of Palwal District of Haryana State, India (28.1487°N, 77.3320°E). A total of 1,014 married women of 30–65 years were selected from a Mendelian population of same gene pool to participate in the study. Each of the recruited women was subject to structured interview schedule for collection of data on household composition and reproductive history after obtaining pre-informed written consent. Since menopause itself is a risk factor for adverse cardiovascular outcomes, women with menopause (both natural and surgical) ($N = 447$) were excluded for the present study. Five women in perimenopausal state were also excluded. Of the 1,009 recruited females, 562 women were in the premenopausal category with an age of 30–46 years. These women were categorized into four groups: (i) Group A—absolute cases with history of miscarriage and MeS, (ii) Group B—absolute controls without any history of miscarriage and MeS, (iii) Group C—cases with MeS but lack any history of miscarriage, and (iv) Group D—cases with history of miscarriage but lack MeS. Age-matched women were selected from each group resulting into a total sample size of $N = 80$ for the present study (20 each group). Any woman who had experienced one or more spontaneous pregnancy loss or abortion till the second trimester of the pregnancy was grouped in the case of history of miscarriage (Group D). Woman was classified with MeS (Group C) following guidelines of NCEP-ATP (III) 2005 Revised Guidelines (*Huang, 2009*), that is, a case of MeS shows the presence of a minimum of three abnormalities

out of waist circumference, fasting glucose, triglyceride levels, HDL levels, and blood pressure. Exclusion criteria included menopause, established CVDs (like myocardial infarction, angina, etc.), and medication for any disease/infection in the past 1 month (Table S1).

The blood samples from twenty women (subjects) from Group A (absolute cases) were collected. Similarly, age-matched subjects were selected from the other three groups (groups B, C, and D) and 60 blood samples were collected. A total of 80 blood samples analyzed for the present study. Intravenous blood samples (5 ml) were collected in evacuated tubes with and without EDTA (2.5 ml each) from every participant after overnight fasting, and the samples were transported on ice to the Molecular Anthropology Laboratory, Department of Anthropology, University of Delhi within 2 h of collection and processed for further analysis. Plasma was separated from evacuated tubes with EDTA after centrifugation at $3000 \times g$ for 10 min. Serum was separated from evacuated tubes without EDTA after centrifugation at $3000 \times g$ for 10 min. Plasma and serum samples were aliquoted for single use (to avoid repeated freezing-thawing of the sample) and stored at $-80$ °C till further use. Serum sample was characterized for biochemical parameters (fasting glucose, total cholesterol or TC, triglycerides or TG, high-density lipoprotein-cholesterol or HDL-C) using enzymatic assay by spectrophotometry within 24 h of sample collection. The protein content of plasma samples was determined by Bradford's dye-binding method using bovine serum albumin (BSA) as standard (Bradford, 1976).

## Two-Dimensional gel electrophoresis and analysis of gel images

To analyze differential protein expression in different case groups and to discern the proteomic link between the history of miscarriage and MeS, plasma proteins of different case groups were subjected to 2-dimensional gel electrophoresis. The protein profiles of case groups A, C, and D were compared with group B. Each protein sample (50 μg) was mixed with rehydration buffer (7M urea, 2M thiourea, 2% CHAPS containing 40 mM dithiothreitol or DTT and 0.2% ampholyte, Sigma-Aldrich, St. Louis, MO, USA). Isoelectric focusing (IEF) was carried out using Protean IEF cell (Bio-Rad, Hercules, CA, USA) as per manufacturer's protocol. Proteins were separated in the first dimension using IPG strips (immobilized pH gradient strips, pH 4–7, seven cm, Bio-Rad, Hercules, CA, USA). The IEF conditions used were: 200 V for 1 h followed by 500 V for 1 h and 1,000 V for next 1 h and rest of the focusing at 8,000 V for 5 h for a total of 13,000 VH (Bio-Rad, Hercules, CA, USA). After completing the IEF, strips were equilibrated for 20 min each in equilibration buffer (50 mM Tris-HCl, pH 8.8, 6M urea, 4% (w/v) SDS, and 20% (w/v) glycerol) with 10 mg/ml DTT followed by that containing 40 mg/ml iodoacetamide (Sigma-Aldrich, St. Louis, MO, USA). Proteins were further separated in the second dimension using SDS-PAGE (12–5% gel system). The gels were silver stained and analyzed using Alpha Digidoc 1201 software. The densities of protein bands of groups A, C, and D were compared with group B. Any protein band from groups A, C, and D showing a $\geq$1.2-fold change in density than that of group B was considered differentially expressed. The relative intensities of protein spots
were determined, and densitometric results were subjected to cluster analyses (*Saroha et al., 2012*).

## In-gel digestion of proteins

To extract the protein from silver-stained gels, the gel blocks containing protein spots were carefully excised and trypsinized using trypsin gold following manufacturer's protocol (Promega, USA). The gel blocks were washed with deionized water and kept in destaining solution (0.2% $K_3FeCN_6$ and 0.04% $Na_2S_2O_3$; Sigma-Aldrich, St. Louis, MO, USA) till the stain disappeared. The washing was continued with 100 mM $NH_4HCO_3$ and 100% $CH_3CN$ in 1:1 ratio. The gel blocks were dehydrated with 40 µl of 1:1 acetonitrile (ACN) and deionized water for 15 min and then washed with 40 µl of ACN. The gel blocks were treated with 50 µl of 100 mM ammonium bicarbonate followed by 50 µl ACN (Sigma-Aldrich, St. Louis, MO, USA). The gel blocks were digested with 10 µl digestion buffer containing 0.1 µg trypsin/µl (Promega, USA) for 45 min on ice. The 60 µl digestion buffer without trypsin (1M $CaCl_2$, 1M $NH_4CO_3$) was added again and incubated at 37 °C for overnight. Supernatants were collected and dried in a vacuum centrifuge. The peptides were extracted by varying the concentration of ACN and trifluoroacetic acid (TFA). Gel pieces were sonicated in a water bath for 30 min, dried in speed-vac to remove remaining TFA and ACN and finally stored at −20 °C till further use (*Wang et al., 2012*).

## Matrix-assisted laser desorption ionization-MS/MS

To identify the differentially expressed proteins, the trypsinized peptides were analyzed by MALDI-TOF MS/MS (Applied Biosystems, Foster City, CA, USA; Life Technologies, Carlsbad, CA, USA) system. One µl of peptide solution was mixed with an equal amount of α-cyano hydroxy cinnamic acid (CHCA) matrix solution prepared in 70% $CH_3CN$ and 30% of 0.1% TFA. Minimum S/N (signal/ noise) filter was set at 25 µm with an exclusion list for α-cyano-4-hydroxycinnamic acid (CHCA) matrix peaks for the MS/MS precursor selection. A statistically significant probability of the proteins acceptance was based on Mowse score ($p \leq 0.05$).

## Western blotting

The altered expression of TTR in the plasma of all the three case groups, that is, Group A, C, and D was evaluated using Western blot analysis. Plasma samples of all four groups were separated on 12–5% SDS–PAGE and transferred onto nitrocellulose membrane (Millipore, Burlington, MA, USA). To reduce the chance of non-specific binding, the membrane was blocked using 5% BSA (Sigma-Aldrich, St. Louis, MO, USA) in 1× PBS (phosphate buffered saline, Merck Chemicals, Germany) for 1 h at RT with shaking. Anti-TTR (Abcam, Cambridge, MA, USA) (1:3,000) and GAPDH (1:1,000) were used as a primary antibody for 2 h at RT. The membrane was then incubated with horseradish peroxidase (HRP) conjugated anti-mouse antibody for 1 h with mild shaking. After each step, the membrane was washed with PBST to remove unbound proteins and antibody. Enhanced chemiluminescence assay was used to develop the membrane as per

manufacturer's protocol (G Biosciences, St. Louis, MO, USA) (*Mishra et al., 2018*). Variation in expression of TTR was analyzed by comparing intensities of the signals observed in the absolute case group (Group A) with that of the absolute control (Group B), only MeS (Group C), and only history of miscarriage (Group D).

## ELISA

To evaluate the expression pattern of TTR, ELISA was carried out using plasma samples in all the groups A, B, C, and D ($N = 20$ each). Each well of a 96-well microtiter plate (Thermo Scientific, Waltham, MA, USA) was coated with anti-TTR antibody (ProSci Antibody, Poway, CA, USA) in 1:2,000 dilution with coating buffer (0.01M $Na_2CO_3$ and 0.035M $NaHCO_3$, pH 9.6) and incubated at 4 °C for overnight. Next day, the plate was washed three times with 100 µl PBST (phosphate buffered saline with tween 20) and incubated with 200 µl blocking buffer (2% BSA; Sigma-Aldrich, St. Louis, MO, USA) for 1 h at RT. The blocking buffer was removed, washed and incubated with plasma samples (1:500) for 3 h at RT. After treating with plasma, plates were washed and incubated with primary antibody (anti-TTR). Again, plate was washed and incubated with 100 µl of anti-mouse HRP conjugate secondary antibody for 1 h and developed with ortho phenylene diamine (1 mg/ml in 0.05M citrate buffer and five µl/ml $H_2O_2$; Sigma-Aldrich, St. Louis, MO, USA) for 30 min. The reaction was stopped by adding 2N $H_2SO_4$ to each well and $OD_{492nm}$ was measured using ELISA plate reader (*Biswas et al., 2013*).

## Protein networking analysis

Differentially expressed proteins were functionally annotated, and their network of interactions was predicted using bioinformatics web tools. Blast2Go bioinformatics platform (*Conesa & Götz, 2008*) was used for functional annotation of 18 proteins differentially expressed in different case groups. To understand the possible biological processes linking the history of miscarriage with MeS, protein-protein interactions were examined: (i) involving all the 18 differentially expressed proteins, and (ii) considering only the seven proteins (HP proteins, ApoA1, transferrin (TF), Serpin38, and SYNE 1) commonly dysregulated in all the three case groups and TTR (involved only in absolute cases exclusively). Functional association among differentially expressed proteins in all case groups was revealed by networking analysis using STRING webtool (Search Tool for the Retrieval of Interacting Genes/Proteins ver. 10.0). The analyses were based on the known and predicted protein-protein interactions (physical and functional) available in the database. The database houses the information derived from sources, such as interactions known from high-throughput experiments, co-expression analysis, and previously available literature, and interactions predicted from computational analyses of genomic characters and gene/protein functions based on orthologs and interologs. The protein–protein interaction with confidence score >0.700 was considered. The confidence score was calculated by weighing and integrating the data for all protein interactions. Colored nodes represent direct interaction, white nodes represent the absence of interaction, whereas the thickness of lines signifies the strength of associations (*Bag et al., 2014*).

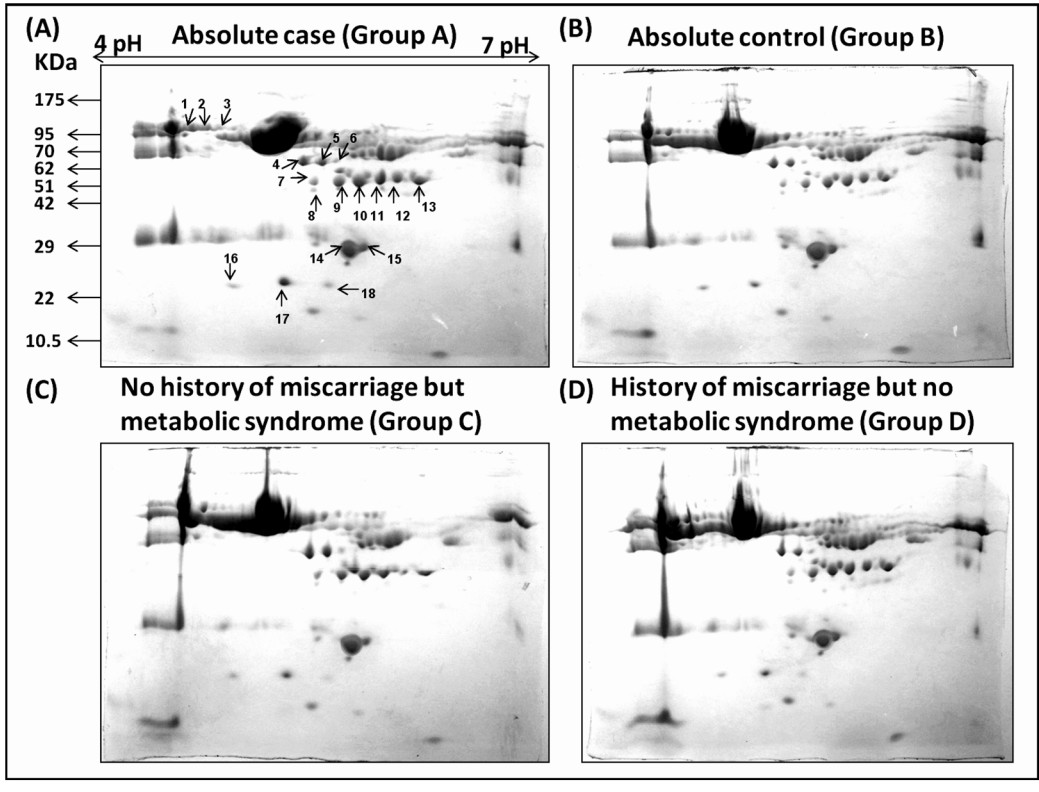

**Figure 1 Comparative 2-DE analysis of plasma proteins.** Silver stained two-Dimensional gel electro-phoretic profile of plasma proteins from four different groups ((A): absolute cases with the history of miscarriage and metabolic syndrome (MeS) (Gr A); (B): absolute controls without history of miscarriage and MeS (Gr B); (C): MeS alone (Gr C); and (D): history of miscarriage alone (Gr D)). Please note the differential expression of 18 proteins among absolute cases, history of miscarriage, MeS and comparing with absolute controls. An equal amount (50 μg) of plasma proteins were separated by four to seven pH range IPG strip (seven cm) followed by SDS-PAGE (12%) and silver stained. Differentially expressed protein spots were marked with numbers.

## Statistical analyses

One-way ANOVA and principal component analysis was performed using SPSS ver. 16.0 for Windows (SPSS Inc., Chicago, IL, USA) and GraphPad Prism 7.01 for Windows (GraphPad Software, La Jolla, CA, USA). Statistical significance was tested at $p < 0.05$. Cluster analysis was done using Multiple Experiment Viewer software (MeV, Boston, MA, USA).

## RESULTS

### Identification of differentially expressed proteins in only miscarriage, only metabolic syndrome, and absolute cases

Densitometric analyses of 2-D gel electrophoretic profiles of plasma proteins of different case groups revealed 18 differentially expressed proteins in groups A, C, and D (Fig. 1). MALDI-TOF/MS-MS analysis revealed that eighteen differentially expressed proteins (Table 1) play significant roles in lipid metabolism, ion transport, and binding processes in the cell. Most of the proteins (15/18) were upregulated in case groups

**Table 1 Details of differentially expressed proteins identified by MALDI-TOF/MS-MS.**

| Spot no. | Accession number | Protein name | MOWSE score | M.W. (Da)/PI calculated | Peptide matched |
|---|---|---|---|---|---|
| 1 | gi\|377656487 | Chain B, transferrin-binding protein B | 173 | 77,301/6.29 | 29 |
| 2 | gi\|380258836 | Chain A, serum transferrin | 141 | 77,299/6.29 | 21 |
| 3 | gi\|110590599 | Transferrin | 60 | 76,988/6.29 | 15 |
| 4 | gi\|256032689 | Chain C, recombinant gamma fibrinogen | 51 | 35,511/5.2 | 17 |
| 5 | gi\|119625326 | Fibrinogen gamma chain | 148 | 47,971/5.2 | 29 |
| 6 | gi\|223170 | Fibrinogen gamma | 132 | 46,823/5.2 | 14 |
| 7 | gi\|764091351 | Chain C, haptoglobin–hemoglobin receptor | 114 | 29,079/5.85 | 15 |
| 8 | gi\|47124562 | HP protein | 92 | 31,647/7.55 | 20 |
| 9 | gi\|73623033 | Serpin 38 isoform b | 52 | 27,940/5.28 | 12 |
| 10 | gi\|6330957 | Synaptic nuclear envelope protein 1 | 49 | 15,745/5.1 | 35 |
| 11 | gi\|47124562 | HP protein | 69 | 31,647/7.55 | 15 |
| 12 | gi\|119600328 | Isoform CRA_a | 54 | 5,141/5.32 | 8 |
| 13 | gi\|78174390 | HP protein | 94 | 38,868/6.13 | 18 |
| 14 | gi\|90108664 | Human apolipoprotein A1 | 219 | 28,061/5.56 | 37 |
| 15 | gi\|90108664 | Human apolipoprotein A1 | 100 | 28,061/5.56 | 29 |
| 16 | gi\|2979530 | Fibroblast growth factor | 50 | 14,685/11.45 | 9 |
| 17 | gi\|226492892 | Coiled-coil domain-containing protein 105 | 52 | 57,386/9.12 | 19 |
| 18 | gi\|2098257 | Transthyretin | 126 | 13,886/5.49 | 13 |

**Note:**
M.W, molecular weight; HP, haptoglobin; PI, isoelectric point.

(A, C, and D), while a few (3/18) showed down-regulation, that is, apolipoprotein A, chain B TF binding protein and chain A TF.

## Clustering of differentially expressed proteins based on their expression

In cluster analysis, the 18 proteins were segregated based on the expression pattern (Fig. 2A) and formed five clusters (Fig. 2B). Different isoforms of ApoA1 showed different expression patterns, and therefore formed different clusters. In cases with only MeS (Group C), one of the isoforms (S15) was upregulated (cluster 2), while the other isoform (S14) was downregulated (cluster 4). Similarly, the expression pattern of two isoforms of chain A, and TF (S2 and S3) also formed distinct clusters (cluster 2 and 3). It may be noted that ApoA1 and chain A, and TF are the major components of the vital cellular processes, that is, lipid metabolism and iron transport, respectively. The distinct functions of these isoforms in different case groups need further analyses. Mostly the proteins were involved in biological regulation and cellular processes followed by the proteins involved in metabolic processes (Fig. 3).

**Figure 2 Cluster analysis based on expression patterns of the eighteen differentially expressed proteins.** (A) The SOTA cluster tree illustrates differential expression of proteins in case groups. (B) The 18 differentially expressed proteins were grouped into five clusters based on their expression profiles. The mean of expression profile is marked in pink for each cluster, whereas expression profile of each individual protein in the cluster is represented by gray lines. Gr A represents absolute cases, Gr C represents metabolic syndrome, and Gr D represents the history of miscarriage.

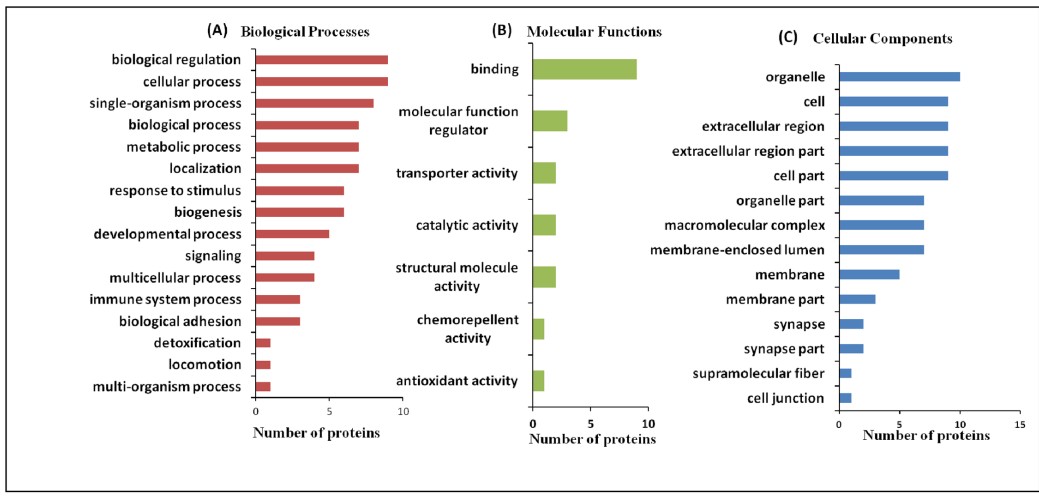

**Figure 3 Functional classification of differentially expressed proteins identified in three case groups.** (A) The biological processes, (B) molecular functions, and (C) cellular components regulated by differentially expressed proteins by Gene Ontology search.

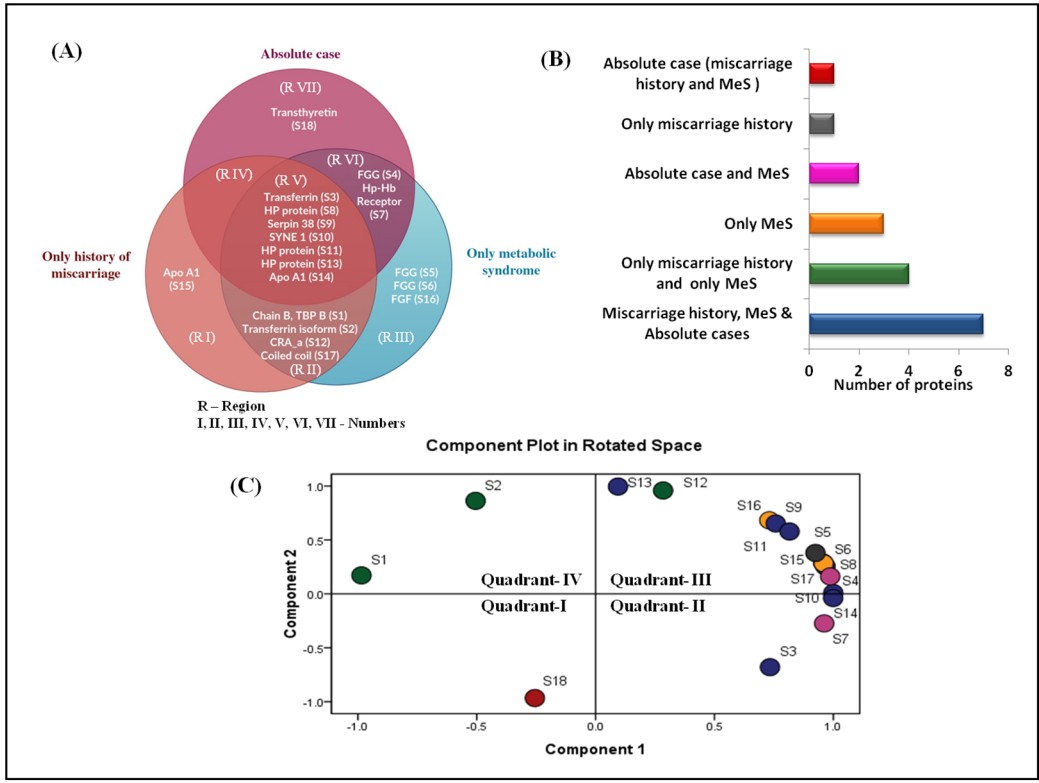

**Figure 4 Venn diagram and PCA analysis of differentially expressed proteins.** (A) Venn diagram showing the distribution of differential expressed proteins among the three case groups. Region V shows seven proteins (transferrin, three different forms of HP proteins, SYNE protein 1, serpin38, apoA1) commonly dysregulated at the intersection of three case groups; Region VII shows exclusive dysregulation of TTR in absolute cases; Region II shows four proteins (chain B transferrin binding protein, transferrin, CRA_a; coiled coil domain) commonly dysregulated at the intersection of cases with miscarriage history, and with MeS; (B) Number of differentially expressed proteins in different pathophysiological states; (C) PCA biplots based on differentially expressed proteins in all three case groups. Protein expressed in the only absolute cases was represented by a red dot in scatter plot whereas green, yellow, pink, gray, and blue dots belonged to patients having miscarriages + MeS, only MeS, absolute case + MeS, only miscarriages and all four groups, respectively.

The Venn diagram shows the distribution of differentially expressed proteins among three case groups (Figs. 4A and 4B). Of the 18 differentially expressed proteins, 11 proteins were shared by the groups C (MeS) and group D (history of miscarriage) that include, ApoA1, TF (two isoforms), chain B TF binding protein B, HP proteins (three isoforms), SYNE protein 1, serpin38, CRA_a and coiled-coil domain protein (regions II and V). Of these, seven proteins were also shared by Group A case (region V). Cases from group C and group D shared only four dysregulated proteins (region II), whereas group A and group C cases shared only two proteins (region VI). Interestingly, group A and group D cases lacked any shared protein (region IV). The cases from group A (region VII) and group D (region I) showed one group-specific protein, whereas cases of group C (region III) showed three group-specific proteins. Transthyretin (TTR, S1) showed upregulation in group A (absolute cases) exclusively and occupied

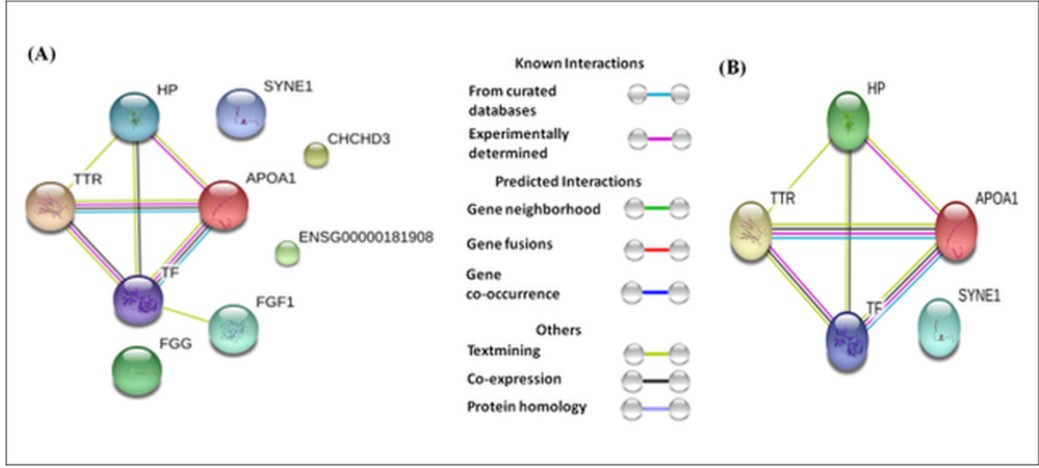

**Figure 5 Protein-protein interaction network based on STRING analysis.** The protein-protein interactions among 18 differentially expressed proteins identified in the present study (A), and seven proteins commonly dysregulated in three case groups and TTR (B). The protein-protein interactions with confidence score >0.700 were considered.

distinct quadrant I in the PCA plot (Fig. 4C), so TTR was chosen for further verification in all the samples.

## Analyses of interactions among commonly dysregulated proteins in the case groups

The differentially expressed proteins showed varying levels of interactions. For example, TTR, TF, ApoA1, and HP showed strong interactions with each other (Fig. 5A), whereas, FGF1 showed a weak interaction with TF. The interaction among TTR, TF, ApoA1, and HP remained strong whether we analyzed all eighteen differentially expressed proteins, or only the seven proteins (HP proteins, ApoA1, TF, Serpin38, and SYNE 1) commonly dysregulated in the three case groups and TTR (involved only in absolute cases exclusively) (Fig. 5).

## Validation of higher expression of transthyretin in absolute cases

ELISA and Western blot analyses confirmed upregulated expression of TTR in the plasma of absolute cases with both history of miscarriage and MeS. As compared with absolute control or group B ($n = 20$), TTR showed significant upregulation in absolute cases or group A, (2.5-fold, $n = 20$) only MeS cases or group C (2.23-fold, $n = 20$) and only miscarriage cases or group D (4.1-fold, $n = 20$) (Fig. 6C). Similarly, the densitometric analysis of Western blots validated a significantly higher expression of TTR in absolute cases (1.56-fold), only MeS cases (1.33-fold) and only history of miscarriage (2.21-fold) as compared with absolute control (Figs. 6A and 6B).

## DISCUSSION

A history of miscarriage makes woman susceptible to metabolic syndrome (MeS); however, the molecular evidence linking the two health conditions is lacking. Proteomics coupled with protein-network analyses have emerged as powerful tools to predict

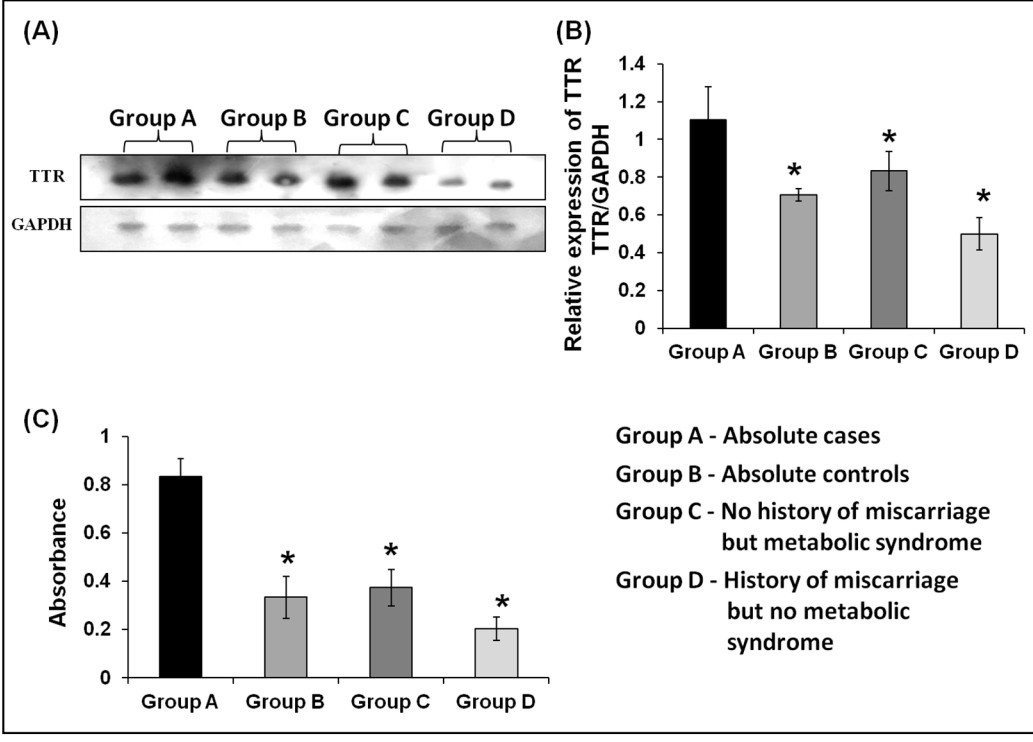

**Figure 6 Validation of expression of TTR in three case groups using Western blot analysis and ELISA.** (A and B) The Western blots analysis revealed 1.56-, 1.33-, and 2.21-fold higher expression of TTR in absolute cases as compared to absolute controls ($p < 0.001$), only MeS ($p < 0.03$) and only history of miscarriage ($p < 0.002$), respectively. (C) ELISA analysis of TTR showed 2.50-, 2.23-, and 4.10-fold elevated expression in absolute cases ($n = 20$) as compared to absolute control ($p < 0.0001$, $n = 20$), only metabolic syndrome ($p < 0.0001$, $n = 20$) and only miscarriages ($p < 0.0001$, $n = 20$), respectively. "*" denotes significant difference in absolute cases as compared to absolute controls, only MeS and only history of miscarriage in Independent $t$-test.

disease-causing genes/proteins, to reveal disease-related biomolecular subnetworks/pathways, and to develop new criteria for disease classification. We used proteomics-based approach to identify the molecular marker (if any) to explain the possible link between the history of miscarriage and MeS. We also developed a protein-protein network to propose a hypothesis to predict the risk for onset of CVDs among women with history of miscarriage and MeS. We recruited women from a Mendelian population of same gene pool so that the study restricts the influence of major confounders (genetic, dietary, cultural, and lifestyle) on proteomic analysis (*Goh et al., 2007*). Also, recruitment of naïve individuals who are not on medication/interventional drugs gives clear picture of the proteome profile in the selected groups. We report 18 differentially regulated proteins among different case groups and showed strong interactions among TTR, TF, ApoA1, and HP.

A dysregulation of proteins of signaling pathways and ion-transport indicate severe physiological turbulence in the cases of a history of miscarriage (group D) and MeS (group C) alone. For example, in both groups C and D, CRA_showed upregulation, and chain B TF-binding protein B and chain A, serum TF showed down-regulation, while

coiled-coil domain protein showed downregulation in group D but upregulation in group C (Fig. 2). A dysregulation of proteins involved in ATP binding, signal transduction and iron transport in a cell has also been reported in several metabolic and neurodegenerative disorders (*Leitner & Connor, 2012*; *Zhao et al., 2015*). Similar to our observations, among cases with miscarriage history and/or MeS, ApoA1, HP, TF, fibrinogen, and TTR have also been reported in independent studies in humans or mouse models (*Rangel-Zúñiga et al., 2015*; *Geyer et al., 2016*; *Hsieh et al., 2016*). Proteomics-based studies have also reported 2–14 proteins associated with preeclampsia and pre-term birth (*Law et al., 2005*; *Metwally et al., 2014*), however to the best of our knowledge, proteomic analyses of the subjects with recurrent miscarrages are lacking. *Metwally et al. (2014)* reported HP and TTR as the key protein among 38 differentially expressed proteins explaining a link between recurrent miscarriages and obesity.

Venn diagram suggested a common dysregulation of seven proteins in all the three case groups (region V, Fig. 4). Increased expression of TF, HP protein (three isoforms), SYNE 1, serpin 38, and ApoA1 in groups A, C, and D provides a better insights into the molecular link between the history of miscarriage and MeS (Table 2). For example, HP serves as an important marker for adiposity and inflammatory conditions in human (*Friedrichs et al., 1995*; *Chiellini et al., 2004*). Increased expression of HP in group A and C could be due to the presence of MeS. However, in Group D women the increased expression of HP suggests that the "stress test" of a female during pregnancy (*Allen, 2016*) resulting in miscarriage may have left some inflammatory signatures, where increasing expression of HP remains one facet. However, the role of haptoglobin (HP) gene polymorphism in better reproductive outcomes can also not be ignored (*Metwally et al., 2014*).

Group C cases with a history of miscarriage showed ApoA1 as a group-specific protein (region I, Fig. 4). In women, the history of miscarriage has been implicated in atherosclerosis (*Sharma & Gulati, 2013*) which in turn is a result of endothelium damage in the arteries due to cholesterol deposition (*Nordestgaard & Varbo, 2014*). As ApoA1 binds with the lipid molecules, specifically HDL-C (*Marcovina & Packard, 2006*), the lower expression of ApoA1 results into the lower level of HDL-C (*Davidson et al., 1996*) leading to catastrophe in the lipid metabolic pathway. The Group D women showed lower level of ApoA1 as compared with those with absolute controls (Fig. 2), which is in concurrence of the findings of *Fanshawe & Ibrahim (2013)*. As women with MeS show low levels of ApoA1 (*Borja et al., 2017*), therefore lowering of ApoA1 isoforms in cases the history of miscarriage in our study imply a physiological interaction among common proteins leading to common protein expression patterns in both the diseases.

Transferrin, a carrier protein synthesized in the liver, acts as a prooxidant molecule as it inhibits iron-dependent hydroxyl radical (OH°) formation from $H_2O_2$ (*Halliwel & Gutteridge, 1989*) and thus (*Memişoğulları & Bakan, 2004*). MeS, a metainflammation state, develops due to metabolic-triggered inflammation (*Monteiro & Azevedo, 2010*) causing oxidative stress (*Pasarica et al., 2009*) and thus becomes deleterious for vascular functions (*Esposito & Giugliano, 2004*). Women with history of miscarriages and preterm births show high inflammation (*Stephenson, 2007*;

**Table 2 Differential expression (fold change) of the identified proteins in three case groups (group A, C, and D) in comparison to the absolute controls (group B).**

| Protein spots | Protein names | Absolute case (Group A) fold change | Metabolic syndrome (Group C) fold change | History of Miscarriage (Group D) fold change | p-value |
|---|---|---|---|---|---|
| S1 | Chain B, transferrin binding protein B | – | −2.32 | −1.48 | 0.651 |
| S2 | Chain A, serum transferrin | – | −1.24 | −1.76 | 0.721 |
| S3 | Transferrin | 1.31 | 1.47 | −1.22 | 0.653 |
| S4 | Chain C, recombinant gamma N308k fibrinogen | −1.22 | 1.58 | – | 0.464 |
| S5 | Fibrinogen gamma chain, isoform CRA_o | – | – | – | 0.767 |
| S6 | Fibrinogen gamma | – | 1.45 | – | 0.079 |
| S7 | Chain C, haptoglobin–hemoglobin Receptor | 1.2 | 1.38 | – | 0.641 |
| S8 | HP protein | 1.29 | 1.53 | 1.26 | 0.315 |
| S9 | Serpin 38 isoform b | 1.74 | 1.93 | 1.86 | 0.081 |
| S10 | Synaptic nuclear envelope protein 1 | 1.47 | 1.66 | 1.37 | 0.229 |
| S11 | HP protein | 1.43 | 1.77 | 1.56 | 0.101 |
| S12 | Isoform CRA_a | – | 1.39 | 1.41 | 0.425 |
| S13 | HP protein | 1.35 | | 1.68 | 0.433 |
| S14 | Human apolipoprotein A-1 | −1.32 | −1.64 | – | 0.968 |
| S15 | Human apolipoprotein A-1 | – | – | −1.47 | 0.934 |
| S16 | Fibroblast growth factor | – | 1.25 | – | 0.982 |
| S17 | Coiled-coil domain-containing protein 105 | – | – | −1.28 | 0.937 |
| S18 | Transthyretin | 2.31 | – | – | 0.020 |

**Note:**
S, protein spot number; –, decreased expression.

*Bukowski et al., 2017*). Therefore, an elevated level of TF signifies a biological response to counteract inflammation. Similarly, serpin 38, another anti-inflammatory molecule also showed an increased expression in three case groups with an increasing pattern from absolute cases > history of miscarriages > MeS group.

The groups C and D cases exclusively shared four proteins (region II) (Fig. 4) suggesting both history of miscarriage and MeS may cause CVDs through independent pathways. PCA analyses also support our views as two proteins dysregulated in region II (S1: chainB TF-binding protein, S2: chainA TF) together form a distinct cluster (quadrant IV). Such clustering may be the cause or an effect of the additional independent pathways

operating in both the diseases, but this also suggests that the metabolic signatures of MeS may be common to the history of miscarriage in women.

Fibroblast growth factor (FGF) and isoforms of Fibrinogen gamma chain (FGG) showed differential expression exclusively in MeS cases in our study (region-III, Fig. 4A), which have also been implicated in CVDs. The patients of CVDs show high levels of FGG. High level of FGG has been positively with traditional risk factors for CVD (BMI, TG, fasting glucose) but has been negatively associated with HDL-C (Alexander, 2012). Clinicians have also suggested FGG as one of the target molecules to treat CVDs, inflammation and thrombosis (Farrell, 2004). On the other hand, FGF possesses the angiogenic capacity and shows an inverse relationship with the levels of BMI, TG, fasting glucose, etc., suggesting the multiple metabolic roles of FGF (Domouzoglou et al., 2015; Nies et al., 2015). FGF improves the metabolic profile (low glucose and improve serum lipids) and acts as anti-thrombotic agent. Therefore, FGF expression is the counter-effect of an increase in FGG expression. Further, the exclusive up-regulation of different chains of FGG (S5 and S6, except S4), and FGF (S16) in women with only MeS in contrast with absolute cases suggests that the MeS which is developed independently of miscarriage history differ physiologically from the MeS which is associated with history of miscarriage.

Though the history of miscarriage and MeS serve as independent risk factors for CVDs, the present study showed higher expression of TTR exclusively among women with a history of miscarriage and MeS. Metwally et al. (2014) showed an increased expression of TTR in the endometrium of obese women with recurrent miscarriage and suggested that an ongoing inflammatory reaction in endometrial lining of obese women may contribute to higher risk of miscarriage. So, increased expression of TTR among absolute cases may be the result of the interactions between the commonly dysregulated proteins in three metabolic states, specifically HP, TF, and ApoA1. Also, TTR increase has been implicated in glucose intolerance, obesity, T2DM, cardiomyopathy (Mody et al., 2008; Pandey et al., 2015) and is considered to be one of the target molecules in the therapy of heart failure (Castano, Narotsky & Maurer, 2015). Similarly, differential levels of TTR in serum have been linked various inflammatory diseases including Alzheimer (Velayudhan et al., 2012), osteoarthritis (Akasaki et al., 2015), breast cancer (Villanueva et al., 2006; Römpp et al., 2007), ovarian cancer (Zhang et al., 2004) and hepatocellular carcinomas (Feng et al., 2005). Protein-protein interaction analyses based on either eighteen dysregulated proteins in all case groups or seven commonly dysregulated proteins in three metabolic cases suggest that TTR interacts with HP, ApoA1, and TF. A cross talk between HP, ApoA1, TF, and TTR estabishs link between miscarriage and MeS. To the best of our knowledge, present study indicates the role of TTR in metabolic syndrome vis-à-vis miscarriage for the first time.

## Strengths

The major strength of this study is a collection of samples from a single Mendelian population sharing a common gene pool with a similar lifestyle, physical activity and dietary habits. Such a retrospective study in a single population thereby reduces the effect of genomic variations, dietary habits etc., in different ethnic groups on proteins to the maximum. In a hospital based case-control study, such matching may not be possible.

Since plasma proteome is largely affected by intervention through drugs, insights into the pathways may be obtained through the recruitment of participants who are not under the influence of medication. Therefore, another major strength of the study is the recruitment of naïve participants who were not on any drug/medical prescription. The study also involved precise phenotyping by cross-validation of reproductive history through repeated multiple-questions involving "miscarriage."

## SUMMARY

Absolute case group (women with a history of miscarriage and MeS) showed altered peripheral blood proteome profile with an increased expression of TTR. In contrast, HP and TF showed upregulation and ApoA1 showed downregulation in the three case groups. Networking analyses suggest cross talk between the four dysregulated proteins, that is, HP, TF, ApoA1, and TTR as a connecting link between miscarriage history and MeS. We propose the putative role of TTR in an increased risk of metabolic adversities among women with miscarriages and MeS. However, longitudinal follow-up studies with larger sample sizes would further help to demonstrate the involvement of TTR and significance of its cross-talk with other proteins in the occurrence of CVDs in women with a history of miscarriages and MeS.

## ACKNOWLEDGEMENTS

The authors are thankful to Prof. P. K. Ghosh and Prof. V. R. Rao for their scientific inputs to the project. The authors also acknowledge Dr Priyanka Rani Garg and Shipra Joshi for their help during the fieldwork.

### Funding

The University of Delhi (DU-DST PURSE Phase II) and faculty research provided support to VM and RSS, and CSIR-SRF provided support to SS. The fieldwork for this work was supported under Department of Biotechnology, Ministry of Science and Technology, Government of India sponsored research project (BT/PR14378/MED/30/535/2010).

### Grant Disclosures

The following grant information was disclosed by the authors:
University of Delhi (DU-DST PURSE Phase II) and faculty research.
Department of Biotechnology, Ministry of Science and Technology, Government of India sponsored research project: BT/PR14378/MED/30/535/2010.

### Competing Interests

The authors declare that they have no competing interests.

### Author Contributions

- Saurabh Sharma conceived and designed the experiments, performed the experiments, analyzed the data, prepared figures and/or tables, authored or reviewed drafts of the paper.

- Suniti Yadav conceived and designed the experiments, performed the experiments, analyzed the data, prepared figures and/or tables, authored or reviewed drafts of the paper.
- Ketaki Chandiok analyzed the data, data collection, fieldwork, sample stratification.
- Radhey Shyam Sharma conceived and designed the experiments, contributed reagents/materials/analysis tools, authored or reviewed drafts of the paper, approved the final draft.
- Vandana Mishra conceived and designed the experiments, authored or reviewed drafts of the paper, contributed reagents/materials/analysis tools, approved the final draft.
- Kallur Nava Saraswathy conceived and designed the experiments, authored or reviewed drafts of the paper, approved the final draft.

### Human Ethics

The following information was supplied relating to ethical approvals (i.e., approving body and any reference numbers):

The Department Ethics Committee, Department of Anthropology, University of Delhi granted approval to carry out the study (Ref. No. Anth/2010/455/1).

### Data Availability

The raw data are provided in the Supplemental Files.

### Supplemental Information

Supplemental information for this article can be found online at http://dx.doi.org/10.7717/peerj.6321#supplemental-information.

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
