# Peer review of "Protein signatures linking history of miscarriages and metabolic syndrome: a proteomic study among North Indian women"

_PeerJ, doi:10.7717/peerj.6321_

## Round 0.1 · original submission · Minor Revisions

Dear Authors,

the Reviewers are favorable to the publication of your manuscript in PeerJ after a minor revision.

Please incorporate or discuss the suggested changes and submit a revised version of your manuscript in order to achieve publication .

Best regards

Salvatore Andrea Mastrolia
PeerJ Academic Editor

Reviewer 1 ·

Basic reporting

no comment

Experimental design

1. How many patients were included per group - please provide details.
2. Please statistics analysis of these four group of clinical characteristic of patients, including maternal weight, age, height, BMI, blood pressure, etc. in a Table.
3. Please include the internal standard protein (like β-actin or GAPDH) in Western Blot analysis, by the way, if the number of samples is less than three, how can you caculate the P value?

Validity of the findings

no comment

Reviewer 2 ·

Basic reporting

no comment

Experimental design

no comment

Validity of the findings

no comment

Additional comments

The article entitled " Protein signature linking history of miscarriages and metabolic syndrome: A retrospective proteomic study among North Indian women" want to identify the proteins signatures to understand the connection between the history of miscarriage and metabolic syndrome. These results showing some new findings, however, there are several points need to be improved and revised before further consideration.

>The text must rewriting, because current text is too redundant.
-For example, In Abstract
The sections of Background and Methods must be simple and clearly.
The section of Results do not need repeat the words in the text. It is better to describe the key new findings. What is the “in the three case groups”? What is the “in the two case groups”? Which is the control?

>What are the standards for grouping? Miscarriage and MeS based on what? Must give more information to explanation.

>The quality of 2-DE gel needs to improve, for current image it’s hard to say about the accuracy of the results.

>The number of the differentially expressed proteins is too few; I cannot concluded any significance thighs from Fig3.

>Why the western blot in Fig6 have any control band? The results of validation are not right. In the other hand, the quality of the blot is very low.

>Table 2
This table must give the p-value of each differential expression result.

Reviewer 3 ·

Basic reporting

The manuscript is very well written and clear. The authors have conducted the right experiments to support their hypothesis and cited appropriate references throughout the manuscript.

Experimental design

The experimental design of the study is appropriate and well thought. The methods reported in manuscript is well described.

Validity of the findings

Data reported in manuscript is meaningful. The study reports proper control and replicates.

Additional comments

In this study, entitled "Protein signature linking history of miscarriages and metabolic syndrome: A retrospective proteomic study among North Indian women" authors investigate the presence of protein signatures in premenopausal women cohorts. The authors undertook 2D-gel electrophoresis and mass spectrometry approach to determine the differential expressed proteins in experimental and control cohorts. Based on the data provided authors suggest that, they have identified 18 differentially expressed proteins in experimental groups and they have also validated the up regulated TTR by various protein assays. The study is significant in profiling the protein signatures on human samples of miscarriage and metabolic syndrome, however following things need to be addressed before publication

1: There is no term like single Mendelian Population. Author should change it to “Mendelian population of same gene pool”.

2: In figure 6, it is not clear whether the authors have compared group A with other three groups or three experimental groups are compared with control group (Group B). Figure legend needs restructuring.

3: Did authors follow up on individuals with increased TTR for their cardio vascular parameters? It is important to know if they show such sign of CVD disorder.

4: Discussion section is too long. It can be shorten a bit.

5: While citing fig in manuscript author should follow one consistent pattern of as advised by journal guidelines. Presently, both round brackets (... ) and square brackets [... ] are used.

---

## Round 0.2 · Minor Revisions

Dear Authors,

After the first round of comments your manuscript was assigned minor revisions.

While Reviewer #3 is now favorable to publication of your manuscript after reading its revised version, Reviewer #2 considered their comments were not adequately addressed. Unfortunately they haven't indicated which of their comments were not addressed, so I appreciate you don't have any feedback to go on

Please prepare a point by point rebuttal letter specifically explaining how you have addressed (or not) the comments from Reviewer 2, in order for the Editorial Board to make a final decision on your manuscript.

Best regards

Salvatore Andrea Mastrolia
PeerJ Academic Editor

Reviewer 2 ·

Basic reporting

no comment

Experimental design

no comment

Validity of the findings

no comment

Additional comments

The authors not address all my concern and questions.

Reviewer 3 ·

Basic reporting

No Comments

Experimental design

No Comments

Validity of the findings

No Comments

Additional comments

The authors have done a commendable job in answering this reviewer questions and in revising the manuscript. No more comments.

---

## Round 0.3 · accepted · Accept

Dear Authors,

I would like to compliment with you for the efforts provided in addressing the Reviewer's comments.

Your manuscript has been considered suitable for publication and can be accepted in its current form.

Best regards

Salvatore Andrea Mastrolia
PeerJ Academic Editor

# Reviewer 2 ·

Basic reporting

accept

Experimental design

accept

Validity of the findings

accept

Additional comments

accept